# Protocol: Effects of midazolam on postoperative delirium in elderly patients undergoing spinal surgery: A randomized, double-blind, placebo-controlled non-inferiority trial

Chaoxu Sheng[1,2°], Miao Zhu[1,2°], Liyong Yuan[1,2‡*], Jianlin Wang[1,2], Bo Hu[1,2], Xiaolu Huang[3], He Han[1,2‡*]

1 Department of Anesthesiology, Ningbo No.6 Hospital, Ningbo, Zhejiang, China, 2 Ningbo Clinical Research Center for Orthopedics, Sports Medicine & Rehabilitation, Ningbo, Zhejiang, China, 3 Department of Operating Room, Ningbo Hospital of Integrated Traditional Chinese and Western Medicine, Ningbo, Zhejiang, China

° These authors contributed equally to this work.
‡ These authors also contributed equally to this work.
* yuanl1973@yeah.net (LY), 15968995356@163.com (HH)

## Abstract

### Introduction

Postoperative delirium (POD) is a common complication in elderly patients, linked to prolonged hospitalization, increased morbidities, and mortality. Midazolam, a widely used perioperative benzodiazepine, has controversial associations with POD: some studies suggest risks, while others find no significant link, but evidence quality remains low. This trial explores whether midazolam is non-inferior to placebo in POD incidence among elderly spinal surgery patients.

### Methods and analysis

A single-center, randomized, double-blind, placebo-controlled non-inferiority trial at Ningbo No. 6 Hospital, China, enrolling 692 elderly patients undergoing elective spinal surgery under general anesthesia. Exclusions include emergency surgery, long-term preoperative benzodiazepine use, and severe cognitive impairment. Participants are randomized 1:1 to receive midazolam (2 mg) or placebo (saline) during induction, via sealed envelopes. Blinding is maintained except for the preparing nurse. Primary outcome: POD incidence assessed by Evaluation tool: Confusion Assessment Method (CAM) on postoperative days 1–3. Secondary outcomes include POD details, agitation (Richmond scale), pain (NRS), lab indicators, hospital stay, and complications.

**Data availability statement:** All data supporting the conclusions of this study will be made publicly available in accordance with the PLOS ONE Data Availability Policy. De-identified raw data (including baseline characteristics, intraoperative records, postoperative outcome assessments, and laboratory results), statistical analysis protocols, and variable definition tables will be deposited in the Figshare repository (https://figshare.com/) upon study completion. A persistent Digital Object Identifier (DOI) will be generated to enable permanent access to the deposited materials. To protect patient privacy, all data will be de-identified by removing personal identifiers (e.g., names, ID numbers, and hospital admission numbers) prior to deposition. Access to the data will comply with ethical guidelines, and researchers may use the data for reproducibility and secondary analysis purposes with appropriate citation of this study. The data deposition process will be completed concurrently with the publication of the study results, ensuring timely availability of all supporting materials to enhance research transparency and reproducibility.

**Funding:** This work was supported by the Ningbo Clinical Research Center for Orthopedics, Sports Medicine & Rehabilitation (Grant No.: 2024L004). The peer review comments of the funding body have not been made public, so we are unable to provide specific peer review opinions. There was no additional external funding received for this study.

**Competing interests:** The authors have declared that no competing interests exist.

## Introduction

Postoperative delirium (POD) is a common postoperative complication in elderly patients, characterized by acute and fluctuating impairments in attention and consciousness [1]. The sequelae of POD are multifaceted, including prolonged hospitalization, an elevated risk of other postoperative morbidities, an increased incidence of cognitive dysfunction and dementia, and even mortality [2].

Midazolam, a short-acting benzodiazepine, remains the most commonly used agent in the perioperative setting. It provides effective sedation, reduces anxiety, and induces amnesia; particularly when combined with agents such as propofol, without significantly prolonging recovery or increasing cardiorespiratory risks in healthy adults [3,4]. Additionally, it significantly reduces the incidence of postoperative nausea and vomiting and the need for rescue antiemetics [5–7].

Previous researchers from the intensive care unit indicated that benzodiazepines may increase the incidence of POD, and clinical guidelines suggested avoiding benzodiazepines in older patients [8–10]. Nevertheless, a systematic review and meta-analysis showed that, in the perioperative setting, benzodiazepine use did not increase the incidence of POD [11]. Moreover, investigations specifically focusing on midazolam failed to find a relationship between midazolam and POD [12–15]. However, the authors pointed out that the quality of evidence to date remains very low, due to a lack of randomization or blinding, large sample sizes, and placebo-controlled comparisons to identify the possible harm of benzodiazepines in the surgical population.

As a consequence, we plan to conduct a randomized, double-blind, placebo-controlled non-inferiority trial aimed at comparing the incidence of postoperative delirium (POD) between midazolam and placebo. We hypothesize that midazolam will have a non-inferior effect on the incidence of POD compared to placebo.

## Methods

### Ethics and dissemination

The research protocol and informed consent form of this study have been approved by the Ethics Committee of Ningbo No.6 Hospital (Ethics No.: 2024–99L). The study has been registered at the Chinese Clinical Trial Registry (http://www.chictr.org.cn), with the registration number ChiCTR2400092537. The registration date is November 19, 2024. This study strictly adheres to the Declaration of Helsinki. Personal information will be kept confidential unless authorized. Data will be retained for at least five years.

This protocol adheres to the guidelines outlined in the Standard Protocol Items: Recommendations for Interventional Trials (SPIRIT). The schedule for participant enrollment, interventions, and assessments is depicted in Fig 1, while Fig 2 illustrates the study design.

### Status and timeline

**Participant recruitment:** Recruitment is scheduled to initiate on October 1, 2025, and has not yet commenced. Based on the recruitment plan and projected progress, we anticipate completion by April 1, 2026.

| STUDY PERIOD | | | | | | | |
|---|---|---|---|---|---|---|---|
| | Enrolment | Allocation | Post-allocation | | Follow-up | | |
| **TIMEPOINT** | *1 day before surgery* | *Surgery day* | *Anesthesia induction* | *PACU* | *1 day after surgery* | *2 days after surgery* | *3 days after surgery* |
| **ENROLMENT:** | | | | | | | |
| Eligibility screen | X | | | | | | |
| Informed consent | X | | | | | | |
| Medical history | X | | | | | | |
| Allocation | | X | | | | | |
| **INTERVENTIONS:** | | | | | | | |
| midazolam | | | X | | | | |
| placebo | | | X | | | | |
| **ASSESSMENTS:** | | | | | | | |
| Mini - Mental State Examination | X | | | | | | |
| Confusion Assessment Method | | | | | X | X | X |
| Richmond Agitation - Sedation Scale | | | | X | | | |
| Numerical Rating Scale | X | X | X | X | X | X | X |
| Time from the end of surgery to tracheal extubation | | | | X | | | |
| Adverse event | | X | X | X | X | X | X |

**Fig 1. Schedule of enrolment, interventions and assessments for the trial.**

**Data collection:** Data gathering will conclude concurrently with recruitment. Since our protocol mandates immediate data collection upon enrollment and all essential data can be acquired during this period, the data collection process will end once the final participant is recruited on April 1, 2026.

**Results availability:** Data analysis will commence after collection is finalized. Considering the complexity of the data and the statistical approaches required, results are expected to be available by May 1, 2026.

## Subjects and setting

We plan to enroll 692 patients scheduled for elective spinal surgery under general anesthesia at Ningbo No. 6 Hospital, Ningbo, Zhejiang, China.

**Inclusion criteria**:

1. Aged 65–90 years

2. Scheduled for elective spinal surgery under general anesthesia

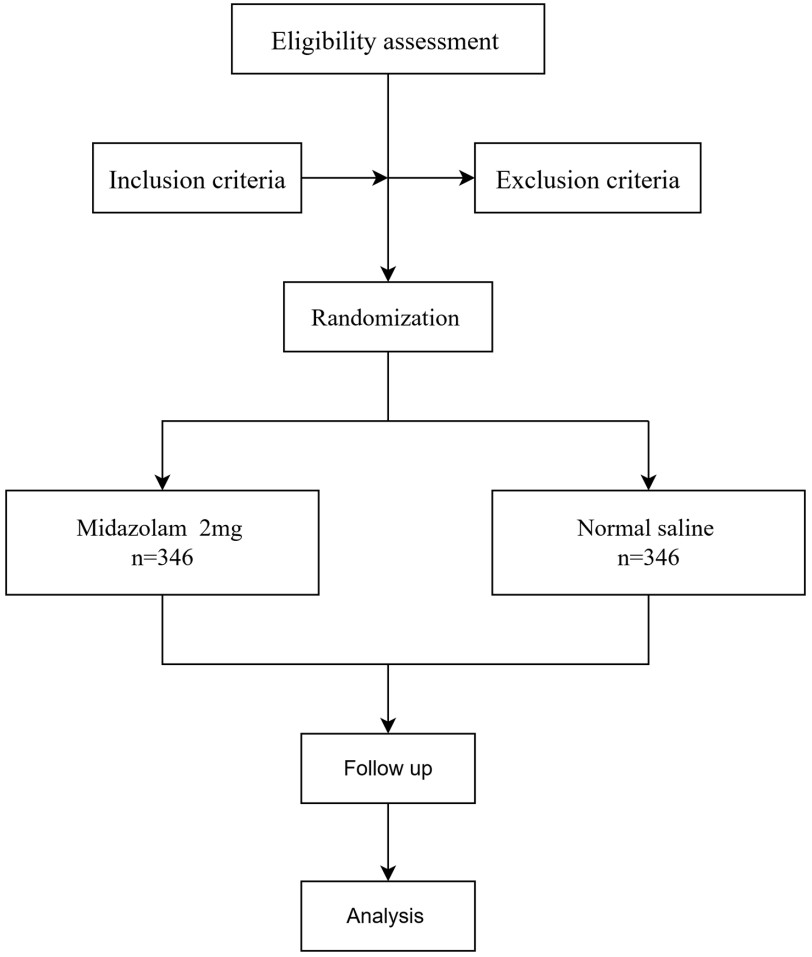

**Fig 2. Flow chart of patient recruitment.**

3. American Society of Anesthesiologists (ASA) physical status I–III

4. Willing to participate and has signed the informed consent form

   **The exclusion criteria will be as follows**:

1. Emergency surgery

2. Patients who need to be transferred to the Intensive Care Unit (ICU) before and/or after surgery

3. Preoperative long-term use of benzodiazepines

4. Diagnosis of cerebrovascular disorders (e.g., stroke, atherosclerotic stenosis of the carotid artery, or transient ischemic attack)

5. No preoperative cognitive function assessment or impaired cognitive function (defined as a preoperative Mini–Mental State Examination [MMSE] score < 18)

6. Patients who cannot be followed up for 3 days after surgery.

7. Patients with moderate-to-severe hearing impairment requiring assistive devices (may hinder verbal interaction for Confusion Assessment Method [CAM]-based POD evaluation)

8. Patients with moderate-to-severe uncorrected visual impairment (may interfere with visual observation for CAM-based POD evaluation)

## Participants' consent

One day prior to the surgical procedure, investigators Miao Zhu/He Han/Chaoxu Sheng will perform an eligibility screening on patients slated for elective spinal surgery the subsequent day. For those patients who satisfy the inclusion criteria, the investigators will disclose the study–related details to the patients and their family members.

The disclosed information encompasses the study's primary and secondary objectives, intricate procedural steps, potential therapeutic benefits, possible adverse events, and corresponding risk–mitigation strategies. In the event that patients express their willingness to partake in the study, either they or their legally authorized representatives are required to sign the informed consent document in triplicate.

## Randomization and blindness

The investigator Bo Hu utilizes the SPSS 25.0.0.1 software to generate 692 random numerical values with a random seed of 12345 (via the "Random Number Generator" module). Odd numbers designated midazolam administration, and even numbers indicated placebo.

Each random number and its corresponding group allocation will be recorded on a small card, which will then be sequentially placed into opaque, sealed envelopes. The envelopes cannot be resealed after opening, allowing for detection of prior tampering. Envelopes will be numbered sequentially from 1 to 692, completing the randomization process for all 692 participants.

The investigator Jianlin Wang is responsible for storing the envelopes, which are kept in a locked filing cabinet. An "Envelope Inventory Record Form" is established to document details including envelope number, preparation date, quantity received into storage, and remaining quantity. Each envelope is clearly labeled with "For single use only; non-reusable after opening" to eliminate any form of envelope reuse. When subjects are formally enrolled, the anesthesiology nurse obtains envelopes from Jianlin Wang in ascending order of envelope number (e.g., the first enrolled subject receives Envelope No. 1, the second receives Envelope No. 2, and so on). At the time of collection, the nurse must complete an "Envelope Distribution and Usage Record Form" by filling in the collector's name, collection time, and the subject's unique ID. Jianlin Wang verifies the information and signs the form to confirm the handover.

In the event that an enrolled subject withdraws from the study due to unforeseen circumstances (e.g., intraoperative need for transfer to the ICU, refusal of further follow-up), their opened envelope (including the internal information card) must still be archived under the "used" category and shall not be retrieved or reused. Subsequent newly enrolled subjects shall continue to use the next "unused envelope" in ascending order (e.g., if Subject No. 5 withdraws, the newly enrolled Subject No. 6 directly uses Envelope No. 6). This ensures each new subject corresponds to a "non-activated unique allocation result" in the original random sequence.

The envelopes are to be opened by the anesthesiology nurse immediately prior to the subject's surgical procedure commencement. After opening the envelope, the anesthesiology nurse must record the opening time, group allocation result, and medication administration time in the "Envelope Distribution and Usage Record Form". The opened envelope (including the internal card) shall be archived as an original study record and shall not be retrieved or reused. Based on the information contained within the envelopes, the anesthesiology nurse formulates the general anesthesia induction

medications. For midazolam (Liyuexi; Jiangsu Enhua Co., Ltd.; 5 mg/ml), 5 mg is prepared, and for the placebo, both are handled using 5 ml syringes. An appropriate amount of normal saline is added to each syringe to dilute the substances to a total volume of 5 ml. A blank label stating "study drug" is affixed to the outside of each syringe.

Only the anesthesiology nurse responsible for the medication preparation on that day is privy to the envelope contents. This nurse is prohibited from participating in other aspects of the study and is strictly forbidden from disclosing the envelope contents to anyone. Other than the anesthesiology nurse, all other individuals involved in the study are kept unaware of the intervention measures. The other investigators are tasked with accurately documenting all peri–operative information of the subjects. The subject recruitment phase of this study concludes upon the utilization of all 692 envelopes.

Regular verification: Every week, Investigator He Han (who is not involved in envelope storage or distribution) and Jianlin Wang jointly verify the "Envelope Inventory Record Form" and "Envelope Distribution and Usage Record Form". They confirm that the numbers of used envelopes are consecutive, the quantity of unused envelopes matches the physical inventory, and there is no skipping of numbers, missing numbers, or reuse. The verification record must be signed by both parties to confirm.

Interim inspection: When 50% of subjects have been recruited (approximately 346 cases), a designated member of the Data Monitoring Committee (DMC) will conduct a special inspection of the "envelope management process". Key verification items include: (1) the correspondence between the original random sequence and envelope numbers; (2) the integrity of archived used envelopes (e.g., no damage or loss); (3) the standardization of filling in the "Envelope Distribution and Usage Record Form" (e.g., no missing entries or inconsistencies). After the inspection, an "Interim Inspection Report on Envelope Management" will be issued. Any identified issues must be promptly rectified, and the rectification process and results must be documented for future reference.

Unblinding is permitted only when it is absolutely essential to know whether midazolam has been administered for the further treatment of the subject. The unblinding process is to be carried out by the investigator (Liyong Yuan), while other investigators are expected to maintain the blinded state to the greatest extent possible.

## Anesthesia management

**General anesthesia induction.** Subjects sign the informed consent form a day before surgery, unaware of their assigned study group.

On the elective surgery day, in the ward, subjects prepare for surgery per the standard protocol without preoperative medications. In the operating room, essential monitoring starts: body temperature, electrocardiogram, blood pressure (invasive or non–invasive), and oxygen saturation (blood oxygen concentration tested if needed). Once results are normal, a peripheral or central vein is accessed.

The research team and anesthesiologists jointly induce general anesthesia. The induction medications include: 2 ml of the study agent, propofol (≈1.5 mg/kg), fentanyl (3 μg/kg) or sufentanil (0.3 μg/kg), rocuronium bromide (1 mg/kg) or cis–atracurium besylate (0.3 mg/kg), and atropine (≈0.25 mg, used as patient–condition–dependent). Dosages are determined by the anesthesiologist [16].

During induction, researchers inject the study reagent. Other medications are administered as the on–duty anesthesiologist requests. After induction, researchers ask the anesthesiologist to record all perioperative details on the electronic anesthesia record sheet and then leave. After surgery, researchers extract needed information from the same electronic record sheet and verify with the anesthesiologist if necessary.

**General anesthesia maintenance.** Sedation is maintained via the utilization of a combination of propofol and sevoflurane, or the exclusive administration of either agent. Analgesia is sustained through the intermittent intravenous bolus injection of fentanyl or sufentanil, or the continuous intravenous infusion of remifentanil. Muscular relaxation is achieved through the administration of rocuronium bromide or cis–atracurium.

The specific pharmacological agents and their respective dosages are meticulously determined by the attending anesthesiologist, who takes into account the patient's physiological status, surgical requirements, and potential drug interactions. Moreover, the selection of vasoactive medications is also at the sole discretion of the on–site anesthesiologist, who tailors the choice to the patient's hemodynamic stability and intraoperative needs.

**General anesthesia recovery.** Upon completion of the surgical procedure, all patients will be transferred to the Post–Anesthesia Care Unit (PACU). In the event that a patient, due to unstable condition or other circumstances, requires immediate postoperative transfer to the ICU, including those who are transferred directly to the ICU upon the conclusion of the surgery and those who are first transferred to the PACU but later require transfer to the ICU due to unstable condition or other reasons, these patients will be treated as drop–out subjects. The envelope assigned to the dropped-out subject shall be archived, and subsequent newly enrolled subjects shall use unused, sequentially ordered envelopes.

In the PACU, a ventilator is utilized to maintain normal ventilation and oxygenation of the subjects. When the patient meets the extubation criteria, neostigmine (1–2 mg) and atropine (0.5–1 mg) may or may not be used for antagonism, as determined by the responsible anesthesiologist. Generally, no other medications are administered. In case of special circumstances, they shall be accurately recorded on the PACU record sheet. Postoperatively, the researchers will extract relevant information to determine whether the subject has dropped out.

Before leaving the PACU (specific time: 30 minutes prior to meeting the PACU discharge criteria, and avoiding the 30-minute window immediately after administration of analgesics/sedatives), the researchers (Miao Zhu/He Han/Chaoxu Sheng) will score and record the subjects using the CAM and Richmond Agitation–Sedation Scale. Additionally, the Numerical Rating Scale (NRS) score and other relevant data will also be recorded.

**Postoperative pain control.** A standardized analgesia protocol will be applied to all participants, consisting of an intravenous patient-controlled analgesia (PCA) pump (baseline analgesia) and on-demand rescue analgesia, as follows:

1. **Intravenous PCA pump**

   - **Formulation**: 100 mL solution containing butorphanol 6 mg, granisetron 6 mg, and normal saline (diluent).

   - **Parameters**: Background infusion at 2 mL/h (0.12 mg butorphanol/h); patient-controlled bolus of 1 mL (0.06 mg butorphanol) with a 15-minute lockout.

   - **Timeline**: Initiated in PACU, maintained until postoperative day 3.

2. **On-demand rescue analgesia**

   - **Drug**: Flurbiprofen axetil 50 mg (intravenous push over ≥1 minute).

   - **Trigger**: NRS pain score ≥4, with ≥4 hours since last rescue dose.

   - **Dose limits**: 50 mg per dose; maximum 150 mg/24 hours.

   - **Execution**: Administered by trained ward nurses, synchronized with daily NRS assessments (8:00–9:00 a.m. and 16:00–17:00 p.m.).

   - **Documentation**: Recorded in the electronic medical record (administration time, dose, pre- and post-administration NRS scores).

3. **Compliance monitoring**

Daily review of PCA logs and analgesia records by the investigator to ensure protocol adherence. Off-protocol deviations will be documented and reported.

**Follow-up.** On postoperative days 1, 2, and 3, researchers (Miao Zhu/He Han/Chaoxu Sheng) will follow up with subjects and conduct the following assessments twice daily, with specific time windows of 8:00–9:00 a.m. and 16:00–17:00. The CAM will be used to assess and record delirium, while pain levels will be rated using the NRS.

**Adverse events.** The researchers (Miao Zhu/He Han/Chaoxu Sheng) extract the adverse hemodynamic events during the surgery and in the PACU through the electronic anesthesia record system (medical system). These events are defined as hypertension, hypotension, tachycardia, or bradycardia. The specific criteria are as follows:

Hypertension will be recorded when the mean arterial pressure (MAP) is ≥ 30% higher than the preoperative baseline. Hypotension will be recorded when the MAP is ≥ 30% lower than the preoperative baseline. Tachycardia is defined as a heart rate (HR) ≥ 100 beats per minute, and bradycardia is defined as an HR ≤ 45 bpm.

Meanwhile, the relevant treatment measures and records of drug use are also documented. All the recorded data are managed through Research Manager system ( http://www.medresman.org.cn).

## Data collection and management

Within three days of the perioperative period, the researchers (Miao Zhu/He Han/Chaoxu Sheng) will extract information through perioperative visits and by accessing the electronic anesthesia record system (medical system) and the Hospital Information System (HIS). If the researchers have any doubts about the data, they will promptly verify the information with the patient himself/herself or the attending anesthesiologist.

**Data management.** Patient data will be recorded and managed via Research Manager system (http://www.medresman.org.cn), which was a web-based platform for clinical research data management certified by the Chinese Clinical Trial Registry (http://www.chictr.org.cn).

To ensure data accuracy, reproducibility, and integrity, the following core procedures were established based on the Research Manager system and manual verification mechanisms adopted in the study.

## Double Entry: Minimizing Data Entry Errors

**Core Operations. Personnel Assignment**: Data collection researchers (Miao Zhu/He Han/Chaoxu Sheng) performed the first round of original entry, wherein data were entered into the system within 24 hours and linked to the unique subject ID. Independent data verifier ( Xiaolu Huang) conducted the second round of independent entry within 48 hours, directly inputting data from the original records without referencing the first-round entries.

**Difference Handling**: The system automatically compared the two rounds of entry results and generated a *Difference Report*. Two personnel jointly reviewed the original records (paper-based CAM forms, electronic anesthetic system data) for verification. Discrepancies were adjudicated by a third-party arbitrator (Jianming Chen, member of DMC), and the results were documented and signed before corrections were made.

**Completion Criterion**: Data could proceed to the next step only when the two rounds of entry results were 100% consistent (or any discrepancies had been resolved).

## Range Checks: Eliminating Outliers

**Preset Reasonable Ranges (Based on Protocol Definitions/Clinical Logic).** According to the 17 types of data listed in the "Data collection" section of the protocol, reasonable ranges were set for "continuous variables" and "categorical variables" respectively. Data exceeding these ranges would trigger a system alert.

**Outlier Handling.** After a system alert was triggered, traceability verification (electronic anesthetic records, HIS test reports, paper-based records) was conducted within 24 hours.

For entry errors, the double entry and range check procedures were repeated after correction. For true outliers (e.g., surgical duration exceeding 8 hours for complex operations), a detailed clinical explanation was required and confirmed

 

with a signature. Data that could not be traced were handled as "missing data" (refer to the "Missing Data Handling" section).

### Audit Trails: Ensuring Traceability

**System-Automated Audit Trails (Research Manager System).** An immutable log was automatically generated, recording: operator/time, data entry/modification/deletion details (including pre-modification and post-modification values + reasons), and permission changes.

### Manual Audit Trails

**Envelope Management**: The *Envelope Inventory Record Form* and *Envelope Distribution and Usage Record Form* were checked and signed by two personnel weekly. Opened envelopes were archived by number and inspected by the DMC during the interim analysis;
**Adverse Events**: Excel records were set to revision mode, marking the modifier/time/basis. Dual-person verification of original data was performed monthly with signatures.

### Final Archiving

At the conclusion of the study, electronic data (dataset, audit trails) and paper-based records (difference resolution forms, outlier explanations, signed forms) were submitted to the Ethics Committee. Archived materials were required to pass integrity verification.

### Data collection

1. Basic patient demographic data, including age, gender, height, weight, body mass index (BMI), heart rate (HR), and blood pressure (BP).

2. Preoperative comorbidities, such as hypertension (grade), diabetes, heart disease, liver and kidney diseases, etc.

3. Preoperative cognitive ability (screened by Mini–Mental State Examination, MMSE), living situation (living alone/living with others/in a nursing home).

4. Laboratory tests: hemoglobin (Hb), C–reactive protein (CRP), albumin, high–sensitivity cardiac troponin I (hs–cTnI), N–terminal pro–brain natriuretic peptide (NT–proBNP).

5. Time from the end of surgery to tracheal extubation

6. 12–lead electrocardiogram, echocardiogram, chest X–ray examination.

7. Types of spinal surgery (decompression/discectomy/minimally invasive surgery/fracture).

8. HR, non – invasive BP, invasive BP, oxygen saturation ($SpO_2$).

9. Incidence times of hypertension and hypotension.

10. The number of times hypotension and hypertension were managed by anesthesiologists.

11. The cumulative duration of hypotension or hypertension.

12. The number of episodes of tachycardia and bradycardia.

13. The detailed usage of vasoactive drugs, anesthetics, and other medications during the surgery.

14. The duration of surgery and anesthesia.

15. The volume of intraoperative fluid and blood transfusion (autologous/allogeneic blood), and blood loss.

16. Any adverse reactions and unexpected situations during the peri–operative period.

17. Postoperative complications

## Outcomes

**Primary outcomes.**  Evaluation indicator: Postoperative delirium.

Evaluation tool: Confusion Assessment Method (CAM). CAM is widely used clinically and has been verified to have high sensitivity and specificity.

Evaluation time: The first, second, and third days after surgery.

**Secondary outcomes.**

1. Onset time, duration, and frequency of POD.

2. Incidence of agitation, evaluated using the Richmond Agitation – Sedation Scale.

3. Postoperative Laboratory indicators: CRP, albumin, hs–cTnI, NT - proBNP.

4. Time from the end of surgery to tracheal extubation

5. Pain score, evaluated using the NRS.

6. Length of hospital stay.

7. Postoperative complications.

**Multiplicity handling for secondary outcomes:** Secondary outcomes in this study (including pain, biomarkers, agitation, length of stay, etc.) are exploratory analyses and will not undergo multiplicity adjustment, with details as follows:

1. **Core Principle**

Although repeated testing may increase the risk of false positives, secondary outcomes are only used to supplement descriptions of additional intervention effects and explore potential associations. They do not affect the primary conclusion regarding the primary outcome (postoperative delirium incidence), thus no adjustment is needed.

2. **Key Rationale for No Adjustment**

1. The primary outcome has strictly controlled Type I error through alpha control (one-sided $\alpha = 0.025$), and secondary outcomes do not determine the overall study conclusion;

2. Adjustment would reduce statistical power, potentially masking clinically meaningful trends;

3. Exploratory outcomes aim for "hypothesis generation," which requires validation in subsequent dedicated trials and does not demand strict control in this study.

3. **Reporting Requirements**

Results must clearly label secondary outcomes as "exploratory." Statistical significance should only serve as descriptive reference, with effect sizes and 95% confidence intervals reported alongside p-values.

**Sample size calculation.**  Based on reported data, the incidence of POD in elderly patients undergoing spinal surgery ranges from 0.84% to 24.6% [17–19]. Considering similar study designs [20,21], we set the non-inferiority margin at 9%.

Our clinical data have shown that the incidence of POD in the midazolam group is approximately 20%, comparable to that in the placebo group. Using these data, we calculated the sample size with PASS 15.0.5 (NCSS, LLC, Kaysville, UT, USA). The analysis employed a Z-test (pooled variance) with a one-sided α of 0.025 and a power of 0.8, yielding a required sample size of 311 in each group. Given the potential for subject dropout in clinical trials—particularly among elderly patients undergoing spinal surgery, where a 10% dropout rate is common based on similar studies—we adjusted the sample size to account for this loss. To ensure the intention-to-treat (ITT) analysis retains sufficient statistical power after accounting for dropouts, we expanded the sample size by dividing the initial effective sample size by (1–10% dropout rate). Re-calculation with PASS 15.0.5 (incorporating a 10% equal dropout rate for both groups, consistent with the aforementioned statistical parameters) confirmed a required recruitment of 346 subjects per group, resulting in a total sample size of 692 patients.

**Statistical analysis.** To address the primary non-inferiority objective (evaluating whether midazolam is non-inferior to placebo in POD incidence), clear definitions of analysis populations and their roles are specified as follows:

1. **Primary analysis population: Intention-to-Treat (ITT) population**

The ITT population serves as the core for the primary non-inferiority test, defined as all 692 patients who complete randomization via the sealed envelope procedure, regardless of the following circumstances:

- Whether they actually receive the assigned intervention (e.g., a patient randomized to the midazolam group who does not receive midazolam due to intraoperative hemodynamic instability);

- Whether they complete all primary outcome assessments (i.e., CAM evaluations for POD on postoperative days 1, 2, and 3; e.g., a patient who withdraws from follow-up on postoperative day 2 due to family request);

- Whether they have protocol deviations (e.g., unplanned short-term use of benzodiazepines during PACU stay).

For missing data on the primary outcome (POD incidence, due to incomplete CAM assessments), multiple imputation (MI) will be employed to maintain the balance of baseline characteristics generated by randomization:

- Imputation will be performed using 5 independent cycles, with predictors including baseline variables (age, ASA physical status, preoperative MMSE score), intraoperative factors (surgery duration, blood loss), and available follow-up data;

- Random seed: 67890 was set for MI to ensure reproducibility of random sampling results;

- Convergence verification: Trace plots were used to confirm imputation stability.

Primary analysis method for non-inferiority test: For the primary non-inferiority assessment of POD incidence, the risk difference (RD) will be used as the primary statistical indicator. RD is defined as the absolute difference in POD incidence between the midazolam group and the placebo group, calculated as: RD = POD incidence in the midazolam group (number of POD cases in midazolam group/ total number of patients in midazolam group) – POD incidence in the placebo group (number of POD cases in placebo group/ total number of patients in placebo group).

- The RD and its one-sided 97.5% confidence interval (CI) will be computed using the normal approximation method (Z-test with pooled variance), which is consistent with the sample size calculation approach (one-sided α = 0.025, power = 0.8) described in the "Sample size calculation" section.

- Non-inferiority judgment criterion: Based on the predefined non-inferiority margin of 9% (as stated in "Sample size calculation"), midazolam will be considered non-inferior to placebo in terms of POD incidence if the upper bound of the one-sided 97.5% CI for RD does not exceed 9%.

2. **Secondary analysis population: Per-Protocol (PP) population**

The PP population is used to verify the robustness of the ITT results, defined as patients who strictly adhere to the study protocol and meet all the following criteria:

- Have received the full assigned intervention (midazolam group: 2 mg midazolam during induction; placebo group: 5 ml normal saline; no wrong-group medication administration);

- Complete all primary outcome assessments (valid CAM scores recorded on postoperative days 1, 2, and 3; no missing key assessment time points);

- Have no major protocol violations that could impact outcome interpretation, including:

  - Preoperative long-term benzodiazepine use not identified during screening (retrospectively confirmed via medical records);

  - Unrecorded reasons for ICU transfer (if applicable) or unreported adverse events related to the study drug;

  - Use of prohibited medications (e.g., other sedatives that may affect POD assessment) during the perioperative period.

3. **Relationship between analysis populations and non-inferiority conclusion**

- The primary non-inferiority conclusion will be based on the ITT analysis: This ensures preservation of the randomization-generated group balance, reduces selection bias (avoided by excluding patients with deviations), and enhances the external validity of results (generalizable to elderly spinal surgery patients in routine clinical settings).

- The PP analysis will serve as a supportive check: If the ITT analysis demonstrates non-inferiority and the PP analysis yields consistent results (i.e., the upper bound of the 95% confidence interval for the between-group difference in POD incidence does not exceed the predefined non-inferiority margin of 9%), it will confirm the robustness of the primary finding. If discrepancies exist (e.g., ITT shows non-inferiority but PP does not), a post-hoc analysis will be conducted to explore potential causes (e.g., the proportion of protocol violators in each group, impact of missing data).

4. **Other statistical methods**

The data were processed using IBM SPSS Statistics 25.0.01 (IBM Corp., Armonk, NY, USA). Continuous data were tested for normality using the Shapiro–Wilk test:

- Normally distributed data (e.g., age, BMI, surgery duration) were reported as mean ± standard deviation (SD), with between-group differences analyzed using Student's t–test;

- Non-normally distributed data (e.g., intraoperative blood loss, length of hospital stay) were reported as median (interquartile range, IQR) and analyzed using the Mann–Whitney U test.

Categorical data (e.g., POD incidence, ASA physical status, presence of comorbidities) were reported as counts (percentages), with comparisons using Fisher's exact test (for cell counts <5) or the chi–square ($\chi^2$) test. Multivariate logistic regression will be performed to identify independent influencing factors for POD, with variables including age, preoperative MMSE score, surgery duration, and intraoperative hypotension. A two-tailed P value $<0.05$ was considered statistically significant.

**Data Monitoring Committee.** An independent Data Monitoring Committee (DMC) was established to objectively evaluate trial data per predefined criteria, ensuring participant safety and research scientific rigor. Its operational mechanisms for independence, data access, and decision-making authority are specified below:

1. **Guarantees for DMC Independence**

DMC members are fully independent of the sponsor, investigators, and research institutions with no conflicts of interest. Key safeguards include:

1. **Composition and Qualifications**: The DMC comprises 3 interdisciplinary experts, including 1 geriatric anesthesiologist (from a non-collaborating institution), 1 biostatistician (independent of the study's statistical team), and 1 clinical ethicist (external expert from the Ethics Committee of Ningbo NO.6 Hospital). All members are free from the following conflicts:

   - Employment, consulting relationships, or financial ties with the study sponsor (e.g., midazolam manufacturers);

   - Collaborative research or mentor-student relationships with the research team in the past 3 years;

   - Patents or academic achievements related to the study intervention.

2. **Conflict of Interest Disclosure**: Members must sign a *DMC Member Conflict of Interest Disclosure Form* before enrollment. Any new conflicts arising during the study must be reported within 48 hours, otherwise, members will be disqualified.

2. **Data Access Authority**

   The DMC only accesses de-identified, aggregated data to maintain blinding and security. Specific permissions:

1. **Data Types**:

   ○ Interim data: 50% sample ITT aggregated data (POD incidence, treatment-related SAEs, conditional power) without personal identifiers;

   ○ Safety data: Real-time aggregated SAE reports (e.g., "Group A: 2 respiratory depression cases") without individual participant details.

1. **Blinding**: Data are labeled "Group A/B" (no midazolam/placebo mapping). Only the independent statistician knows group assignments; DMC remains blinded until trial end.

2. **Access Timing**: Data are accessed centrally for interim analysis; SAE reports are submitted within 72 hours of occurrence. No unnecessary access.

3. **Decision-Making and Communication Authority**

   The DMC provides advisory opinions based on predefined criteria (conditional power, safety thresholds) without direct trial intervention. Details:

1. **Decision-Making Scope**:

   ○ Interim decisions: Recommend termination if CP < 20% or SAE difference ≥ 5% (P < 0.01); recommend continuation if CP ≥ 20%, providing written trial continuation/termination advice;

   ○ Emergency safety decisions: Convene emergency meetings for unexpected serious reactions (e.g., midazolam anaphylaxis) to recommend trial pause and cause investigation.

2. **Voting & Documentation**: Decisions require majority vote. Results and discussions are recorded in *DMC Meeting Minutes* with all members' signatures; disagreements are documented with reasons.

3. **Communication Procedures**:

- ○ Recommendations are submitted to Ningbo NO. 6 Hospital Ethics Committee and sponsor; investigators provide written implementation plans within 1 week.

- ○ Communication is restricted to Ethics Committee-approved emails; no private contact. All records are archived for review.

**Interim analysis.** An interim analysis of the primary outcome (POD incidence) will be conducted when 50% of the planned sample size is recruited (i.e., 346 subjects, corresponding to 50% information time), with the primary objectives of: (1) assessing the feasibility of continuing the trial (e.g., whether the trial is unlikely to achieve the non-inferiority endpoint); (2) monitoring for unexpected safety signals; and (3) avoiding unnecessary resource consumption while controlling the overall type I error.

1. **Criteria for early termination or continuation**

The Data Monitoring Committee (DMC), an independent body, will make decisions on trial termination or continuation based on conditional power (CP) (for efficacy-related feasibility) and safety thresholds (for adverse event monitoring), using pre-specified criteria:

(1) **Conditional power (CP)-based efficacy feasibility criteria**

CP quantifies the probability of achieving the primary non-inferiority endpoint at the final analysis, given the accumulated data at the interim point. For this non-inferiority trial (predefined non-inferiority margin: 9%; one-sided $\alpha = 0.025$; power $= 0.8$), the CP will be calculated using the accumulated POD incidence data from the ITT population (with missing data handled via multiple imputation, consistent with the primary analysis plan). The decision criteria are:

- **Termination for futility**: If CP $< 20\%$ (i.e., the probability of confirming non-inferiority at the final analysis is $< 20\%$), the DMC will recommend terminating the trial early, as continuing recruitment is unlikely to yield a positive non-inferiority result and would expose additional subjects to unnecessary intervention.

- **Continuation**: If CP $\geq 20\%$, the trial will continue as planned, as the accumulated data still support a reasonable chance of achieving the primary endpoint.

- **Early confirmation of non-inferiority (rare for non-inferiority trials)**: Given that non-inferiority trials focus on "not being worse" rather than "being better," early termination for positive efficacy will not be considered unless the CP $> 95\%$ (an extremely high threshold) and the upper bound of the one-sided 97.5% CI for the risk difference (RD) of POD incidence is $< 3\%$ (far below the predefined non-inferiority margin of 9%). This strict criterion avoids prematurely overinterpreting the intervention effect and ensures generalizability.

(2) **Safety threshold criteria**

Safety monitoring will focus on treatment-related serious adverse events (SAEs) (e.g., severe respiratory depression, hypersensitivity reactions, or SAEs judged by the DMC to be causally related to midazolam). The DMC will compare the incidence of treatment-related SAEs between the midazolam and placebo groups using Fisher's exact test (for small sample sizes at the interim stage). The decision criteria are:

- **Termination for safety**: If the incidence of treatment-related SAEs in the midazolam group is significantly higher than that in the placebo group (one-sided $P < 0.01$) and the absolute risk difference exceeds 5% (i.e., $\geq 5$ more SAEs per 100 subjects in the midazolam group), the DMC will recommend terminating the trial early to protect subjects from potential harm.

- **Continuation**: If no significant between-group difference in treatment-related SAEs is observed (one-sided $P \geq 0.01$) or the absolute risk difference is $< 5\%$, the trial will continue, with enhanced monitoring of adverse events.

2. **Alpha control and multiplicity adjustment**

To maintain the overall type I error rate (one-sided $\alpha = 0.025$, consistent with the sample size calculation) across the interim and final analyses, an alpha-spending function (O'Brien-Fleming method) will be applied. This method is preferred for non-inferiority trials because it allocates a smaller proportion of the total $\alpha$ to the interim analysis, minimizing the risk of prematurely rejecting the null hypothesis (i.e., falsely concluding non-inferiority) and ensuring the final analysis retains the majority of the $\alpha$ for definitive inference.

**Specific alpha allocation.** Based on the O'Brien-Fleming method and 50% information time at the interim analysis:

• **Interim analysis**: A conservative alpha of one-sided $\alpha_1 = 0.005$ will be used. For the primary non-inferiority test (RD with one-sided 97.5% CI), the interim analysis will only trigger a decision if the upper bound of the 97.5% CI for RD is < 2% (far below the 9% margin) and the statistical test yields P < 0.005. This strict threshold ensures the interim analysis has minimal impact on the total type I error.

• **Final analysis**: The remaining alpha of one-sided $\alpha_2 = 0.020$ will be used ($\alpha_1 + \alpha_2 = 0.025$, maintaining the overall type I error). The final non-inferiority conclusion will be based on whether the upper bound of the 97.5% CI for RD $\leq$ 9% and P < 0.02.

**Multiplicity adjustment for secondary outcomes.** No additional multiplicity adjustment will be applied to secondary outcomes (e.g., POD duration, agitation incidence, lab indicators), as they are exploratory and not used to confirm the primary non-inferiority conclusion. This avoids overly conservative statistical inference for supportive endpoints while focusing type I error control on the primary outcome.

1. **Execution of interim analysis**

• **Blinding maintenance**: The interim analysis will be conducted in a blinded manner. The research team will provide de-identified, aggregated data (e.g., "Group A vs. Group B" POD incidence, SAE counts) to an independent statistician (not involved in subject recruitment or data collection), who will calculate CP and perform safety comparisons. The statistician will only report the results (e.g., "CP = 15%," "SAE incidence difference = 6%, P = 0.008") to the DMC, without disclosing group assignments.

• **Decision documentation**: The DMC will document its discussion, vote, and decision (continue/terminate) in an "Interim Analysis Decision Report," which will be submitted to the Ningbo NO.6 Hospital Ethics Committee for review and archiving. If termination is recommended, the report will include a detailed rationale (e.g., futility or safety concerns) and a plan for subject follow-up (e.g., completing scheduled assessments for enrolled subjects).

**Harms.** If a subject experiences any adverse event after signing the informed consent form but before receiving the study drug intervention, this event shall be reported as unrelated to the study drug. If a subject experiences an adverse event after receiving the study drug intervention, it shall be reported as potentially related to the study drug.

A serious adverse event is defined as an adverse event that, after discussion among researchers, is considered related to midazolam. Such an adverse event includes the following: life–threatening situations, disability, long–term hospitalization, or significant risks determined by the Data Monitoring Committee. Serious adverse events shall be reported to the Ethics Committee.

## Discussion

Benzodiazepines remain widely used in the perioperative period, primarily for their anxiolytic, sedative, and amnestic properties [16,22]. Utilization rates are particularly high in major orthopedic surgeries in the United States, where up to 80% of patients receive benzodiazepines perioperatively [16,23]. Although previous researchers from the intensive care

unit found a relationship between benzodiazepines and POD, and clinical guidelines suggested avoiding benzodiazepines in elderly patients [9,10], a systematic review and meta-analysis showed that, in the perioperative setting, benzodiazepine use did not increase the incidence of POD [11]. Differing from the primary data of previous consensus or guidelines from the ICU [9,14], this study extracts data from inpatient surgeries. It did not find clear evidence regarding the association between benzodiazepines and POD. However, the authors also pointed out that the quality of evidence was very low. Rigorous research is needed to identify the possible harm of benzodiazepines in the surgical population.

A recent randomized crossover trial across 20 medical centers compared restrictive (no benzodiazepines) and liberal (at least 0.03 mg/kg midazolam equivalent) cardiac anesthesia policies in approximately 20,000 patients [24]. Ultimately, policy adherence in both arms exceeded 90%, and the primary analysis showed no significant reduction in the incidence of POD (14.0% vs. 14.9%; adjusted odds ratio [aOR] 0.92). This result is consistent with the conclusion of a systematic review and meta-analysis that perioperative benzodiazepines may not increase the risk of POD [11]. Nonetheless, post-hoc analyses revealed a significantly lower incidence of POD in the restrictive arm (aOR, 0.90; 95% confidence interval [CI], 0.82–0.99) when only policy-adherent patients were included. This suggests that benzodiazepines may still contribute to POD; therefore, further studies are still necessary to confirm the safety of perioperative benzodiazepines.

Another prospective multicenter study investigated noncardiac surgery in 5,663 older patients. The results showed that the most commonly used intraoperative benzodiazepine, midazolam, did not increase the risk of POD [13]. Compared with previous randomized controlled trials (RCTs) of very low quality [11], its observational study design, large sample size, rigorous multivariable adjustment, and standardized delirium assessment qualify it as "moderate" evidence. However, there remains no "high"-quality evidence to date.

Consequently, we aim for this single-center, randomized, double-blind, placebo-controlled non-inferiority trial to provide potentially high-quality evidence for this controversial topic.

## Strengths and limitations of this study

1. **Strength**: The study employs a rigorous randomized, double-blind, placebo-controlled design with strict allocation concealment and blinding procedures, minimizing bias and enhancing the reliability of results.

2. **Strength**: It focuses on a specific population (elderly patients undergoing spinal surgery) and uses standardized assessment tools (e.g., CAM for delirium, NRS for pain), ensuring targeted and consistent outcome measurements.

3. **Limitation**: As a single-center trial, the results may have limited generalizability to other healthcare settings or diverse patient populations.

4. **Limitation**: The non-inferiority margin (9%) is based on previous studies and pilot data, which may introduce subjectivity; further validation of this threshold is needed.

## Supporting information

**S1 File. Reporting checklist for protocol of a clinical trial.**
(DOCX)

**S2 File. Research project submitted to the ethics committee (Chinese).**
(DOC)

**S3 File. Research project submitted to the ethics committee (English).**
(PDF)

## Author contributions

**Conceptualization:** Chaoxu Sheng, Liyong Yuan, He Han.

**Data curation:** Chaoxu Sheng, Jianlin Wang, Bo Hu, He Han.

**Formal analysis:** Chaoxu Sheng.

**Funding acquisition:** Liyong Yuan.

**Investigation:** Chaoxu Sheng, Jianlin Wang, Bo Hu, He Han.

**Methodology:** Chaoxu Sheng, Liyong Yuan, He Han.

**Project administration:** Chaoxu Sheng, Miao Zhu, Liyong Yuan, Jianlin Wang, Bo Hu, Xiaolu Huang, He Han.

**Resources:** Liyong Yuan.

**Validation:** Xiaolu Huang.

**Visualization:** Xiaolu Huang.

**Writing – original draft:** Miao Zhu.

**Writing – review & editing:** Miao Zhu, He Han.

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
