## [Decision Letter · Decision Letter 0]

10 Oct 2025

Dear Dr. He Han,

Thank you for submitting your manuscript to PLOS ONE. After careful consideration, we feel that it has merit but does not fully meet PLOS ONE’s publication criteria as it currently stands. Therefore, we invite you to submit a revised version of the manuscript that addresses the points raised during the review process.

We look forward to receiving your revised manuscript.

Kind regards,

Shweta Rahul Yemul Golhar, MD

Academic Editor

PLOS ONE

Journal Requirements:

“This work was supported by Ningbo Clinical Research Center for Orthopedics, Sports Medicine & Rehabilitation (2024L004).

The peer review comments of the funding body have not been made public, so we are unable to provide specific peer review opinions.”

3. PLOS requires an xlink:type="simple" iD for the corresponding author in Editorial Manager on papers submitted after December 6th, 2016. Please ensure that you have an ORCID iD and that it is validated in Editorial Manager. To do this, go to ‘Update my Information’ (in the upper left-hand corner of the main menu), and click on the Fetch/Validate link next to the ORCID field. This will take you to the ORCID site and allow you to create a new iD or authenticate a pre-existing iD in Editorial Manager.

“This work was supported by Ningbo Clinical Research Center for Orthopedics, Sports Medicine & Rehabilitation (2024L004).”

“This work was supported by Ningbo Clinical Research Center for Orthopedics, Sports Medicine & Rehabilitation (2024L004).”

6. Please include a separate caption for each figure in your manuscript.

Additional Editor Comments (if provided):

Thank you for submitting your research protocol to PLOS One. Upon review of the protocol there are numerous suggestions on statistical analysis and methodology proposed by reviewers. Please review the reviewers questions as noted in the email.

Some additional suggestions are as follows:

Exclude patients with auditory or visual deficiencies as they may be prone for perioperative delirium and difficult to assess.

Specific time of day and number of times per day that assessment will be done.

Specify what post operative pain control modality will be used and ensure its uniformity.

Reviewers' comments:

Reviewer's Responses to Questions

**Comments to the Author**

1. Does the manuscript provide a valid rationale for the proposed study, with clearly identified and justified research questions?

Reviewer #1: Yes

Reviewer #2: Yes

Reviewer #3: Yes

2. Is the protocol technically sound and planned in a manner that will lead to a meaningful outcome and allow testing the stated hypotheses?

Reviewer #1: Yes

Reviewer #2: Partly

Reviewer #3: Yes

3. Is the methodology feasible and described in sufficient detail to allow the work to be replicable?

Reviewer #1: Yes

Reviewer #2: Yes

Reviewer #3: Yes

4. Have the authors described where all data underlying the findings will be made available when the study is complete?

Reviewer #1: Yes

Reviewer #2: Yes

Reviewer #3: Yes

5. Is the manuscript presented in an intelligible fashion and written in standard English?

Reviewer #1: Yes

Reviewer #2: Yes

Reviewer #3: Yes

You may also provide optional suggestions and comments to authors that they might find helpful in planning their study.

Reviewer #1: Thank you for submitting this very clearly written study on an important topic of postoperative delirium after use of benzodiazepines. the protocol is clear, and they have demonstrated proper ethics oversight. I look forward to seeing the results of this study and subgroup analysis.

Reviewer #2: Thank you for the opportunity to review this protocol. I have several comments and suggestions for the authors:

Benefit of Midazolam:

The study focuses primarily on the potential harms of midazolam; however, it remains unclear whether midazolam provides meaningful benefit in reducing anxiety in older adults. A balanced risk–benefit assessment requires an understanding of both harm and benefit. The authors should consider including a measure of postoperative anxiety or patient experience to determine whether midazolam meaningfully improves patient-centered outcomes.

Timing of CAM Assessment:

The protocol does not specify when the CAM is administered (morning, afternoon, or evening) or whether timing is standardized across participants. Since delirium symptoms can fluctuate throughout the day, standardizing the assessment time—or documenting it carefully—is important for ensuring consistency and interpretability. Additionally, if timing is being analyzed as a secondary outcome, clarification is needed on how this will be meaningful without standardized measurement timing.

Loss to Follow-Up:

The authors should clarify how they plan to handle participants lost to follow-up in their analysis. Furthermore, for participants who remain intubated on postoperative day (POD) 1, it would be helpful to describe how these cases will be managed in the assessment of delirium and other outcomes.

Discharge and Population Heterogeneity:

It is unclear whether all participants will remain hospitalized for three days postoperatively. If some are discharged earlier, how will data collection and outcome assessment be handled? Additionally, including both single-level discectomy and multilevel fusion patients in the same cohort may introduce heterogeneity, as these groups differ in surgical magnitude and recovery trajectories. The authors should clarify whether these populations will be analyzed separately.

Reviewer #3: Title: Protocol: Effects of midazolam on postoperative delirium in elderly patients undergoing spinal surgery: a randomized, double-blind, placebo-controlled non-inferiority trial

This is a well-structured, ethically approved, and SPIRIT-compliant trial protocol. The non-inferiority design is appropriate for the study question. The statistical and data-management components are generally sound but require clarification on several points (margin justification, randomization integrity, missing data handling, and consistency of intervention dose)

Major Statistical/Methodological Comments

1. Non-inferiority margin justification: The 9% non-inferiority margin is insufficiently justified. Please explain the clinical rationale (why a 9% higher POD rate is acceptable) and provide supporting references or pilot data. Specify the direction of the hypothesis (upper bound of CI for risk difference).

2. Dose inconsistency: Midazolam dose is reported as 2 mg in the abstract and 5 mg in the Methods. This inconsistency affects both design and power assumptions; correct and harmonize throughout the text.

3. Randomization and allocation concealment: Clarify who generates, stores, and secures the random sequence, and whether allocation concealment is audited independently. The plan to reuse envelopes from withdrawn subjects should be revised to preserve randomization integrity.

4. Analysis population: Specify whether the primary non-inferiority analysis will use intention-to-treat (ITT) or per-protocol (PP) data. Both are expected in non-inferiority frameworks; clarify which is primary.

5. Primary analysis method: Define the statistical test explicitly—e.g., risk difference or odds ratio with one-sided 97.5% CI. For non-inferiority, risk difference is typically preferred for interpretability.

6. Interim analysis and alpha control: Provide criteria for early termination or continuation (e.g., conditional power, safety threshold). Describe whether alpha-spending or multiplicity adjustments will be applied to maintain type I error.

7. Missing data handling: Specify how missing POD outcomes (e.g., due to ICU transfer or death) will be treated. Conservative imputation (e.g., 'missing = event') or multiple imputation methods should be described.

8. Data validation and audit: Data entry via the Research Manager system is appropriate, but please detail procedures for double entry, range checks, and audit trails to ensure reproducibility and data integrity.

9. Multiplicity of secondary outcomes: Numerous secondary outcomes (pain, biomarkers, agitation, LOS, etc.) may increase false positives. Clarify that these are exploratory and will not be adjusted for multiplicity, or specify correction methods.

10. Data Monitoring Committee (DMC): Provide more detail on DMC independence, data access, and decision-making authority. Clarify whether members are independent of the sponsor and investigators.

Minor Comments

• Align the SPIRIT figure and text (the GAD-7 scale appears in Fig 1 but not described in Methods).

• Confirm all software versions (SPSS 19.0, PASS 15.0.5) and ensure reproducibility details (parameters, seed).

• Consider depositing the data and analysis code plan in a named repository per PLOS ONE data policy.

• Proofread for small typographical issues and ensure all references have complete DOIs.

**Do you want your identity to be public for this peer review?** For information about this choice, including consent withdrawal, please see our Privacy Policy

Reviewer #1: No

Reviewer #2: No

Reviewer #3: **Yes:**  Dr Shah-Jalal Sarker

---

## [Author Response · Author response to Decision Letter 1]

10 Nov 2025

Dear Editors and Reviewers,

Thank you for giving me the opportunity to submit a revised draft of my manuscript titled Protocol: Effects of midazolam on postoperative delirium in elderly patients undergoing spinal surgery: a randomized, double-blind, placebo-controlled non-inferiority trial (PONE-D-25-37876) to PLOS ONE.

We appreciate the time and effort that you and the reviewers have dedicated to providing your valuable feedback on my manuscript. We are grateful to the reviewers for their insightful comments on our paper. We have incorporated changes to reflect most of the suggestions provided and have highlighted the revisions within the manuscript.

Journal Requirements:

1. Please ensure that your manuscript meets PLOS ONE's style requirements, including those for file naming. The PLOS ONE style templates can be found at https://journals.plos.org/plosone/s/file?id=wjVg/PLOSOne_formatting_sample_main_body.pdf and https://journals.plos.org/plosone/s/file?id=ba62/PLOSOne_formatting_sample_title_authors_affiliations.pdf.

Response: Thank you for your guidance. We have revised the manuscript in strict accordance with the PLOS ONE style templates. Please feel free to point out any remaining inconsistencies or errors—we greatly appreciate your further feedback.

2. Thank you for stating in your Funding Statement: “This work was supported by Ningbo Clinical Research Center for Orthopedics, Sports Medicine & Rehabilitation (2024L004). The peer review comments of the funding body have not been made public, so we are unable to provide specific peer review opinions.” Please provide an amended statement that declares all the funding or sources of support (whether external or internal to your organization) received during this study, as detailed online in our guide for authors at http://journals.plos.org/plosone/s/submit-now. Please also include the statement “There was no additional external funding received for this study.” in your updated Funding Statement. Please include your amended Funding Statement within your cover letter. We will change the online submission form on your behalf.

Response: Thank you for the instruction. We have updated the Funding Statement as requested, including the required declaration about no additional external funding. The amended statement is included in the cover letter, and we kindly ask for your review.

Response: We confirm that the ORCID iD has been successfully created, validated, and linked to the corresponding author’s profile in Editorial Manager.

4. Thank you for stating the following in the Acknowledgments Section of your manuscript: “This work was supported by Ningbo Clinical Research Center for Orthopedics, Sports Medicine & Rehabilitation (2024L004).” We note that you have provided additional information within the Acknowledgements Section that is not currently declared in your Funding Statement. Please note that funding information should not appear in the Acknowledgments section or other areas of your manuscript. We will only publish funding information present in the Funding Statement section of the online submission form. Please remove any funding-related text from the manuscript and let us know how you would like to update your Funding Statement. Currently, your Funding Statement reads as follows: “This work was supported by Ningbo Clinical Research Center for Orthopedics, Sports Medicine & Rehabilitation (2024L004).” Please include your amended statements within your cover letter; we will change the online submission form on your behalf.

Response: We sincerely appreciate your careful review and valuable guidance on our manuscript. In response to your comment regarding the funding information disclosure, we have completed the required revisions as follows:

1. All funding-related content has been thoroughly removed from the main manuscript, including the Acknowledgments section, to comply with the journal’s policy that funding details should only be presented in the Funding Statement of the online submission form.

2. The funding information in the cover letter has been updated accordingly to align with the revised disclosure requirements.

We kindly request you to review these revisions. Should you require any further adjustments or additional information, please do not hesitate to contact us. Thank you again for your attention and support.

Response: We have removed the ethics statement and Trial Registration Number from the abstract (lines 51-55, 58) while retaining comprehensive ethics-related information in the Methods section, in full compliance with the journal’s requirements.

6. Please include a separate caption for each figure in your manuscript.

Response: We have added a separate, detailed caption for each figure as required (lines 115-116).

Response: Thank you for the reminder. We will carefully review and evaluate all recommended publications to assess their relevance to the study and include appropriate citations where necessary.

Additional Editor Comments (if provided):

Thank you for submitting your research protocol to PLOS One. Upon review of the protocol there are numerous suggestions on statistical analysis and methodology proposed by reviewers. Please review the reviewers’ questions as noted in the email. Some additional suggestions are as follows:

Exclude patients with auditory or visual deficiencies as they may be prone for perioperative delirium and difficult to assess.

Response: Thank you for this valuable suggestion. We agree that patients with auditory or visual deficiencies may face increased risk of perioperative delirium and pose challenges to standardized assessment. We have added this group to the exclusion criteria (lines 147-151) to enhance the validity of outcome measurements.

Specific time of day and number of times per day that assessment will be done.

Response: Thank you for the reminder. We have clearly specified the exact timing and frequency of postoperative assessments in lines 281-283 and 288-290 to ensure consistency across all participants.

Specify what post operative pain control modality will be used and ensure its uniformity.

Response: Thank you for your attention to the postoperative analgesia protocol. To ensure the uniformity of analgesic management and avoid confounding effects on study outcomes, we have implemented a standardized multimodal analgesia scheme for all participants, as follows:

A standardized analgesia protocol will be applied to all participants, consisting of an intravenous patient-controlled analgesia (PCA) pump (baseline analgesia) and on-demand rescue analgesia, as follows:

1. Intravenous PCA pump

• Formulation: 100 mL solution containing butorphanol 6 mg, granisetron 6 mg, and normal saline (diluent).

• Parameters: Background infusion at 2 mL/h (0.12 mg butorphanol/h); patient-controlled bolus of 1 mL (0.06 mg butorphanol) with a 15-minute lockout.

• Timeline: Initiated in PACU, maintained until postoperative day 3.

2. On-demand rescue analgesia

• Drug: Flurbiprofen axetil 50 mg (intravenous push over ≥1 minute).

• Trigger: NRS pain score ≥4, with ≥4 hours since last rescue dose.

• Dose limits: 50 mg per dose; maximum 150 mg/24 hours.

• Execution: Administered by trained ward nurses, synchronized with daily NRS assessments (8:00–9:00 a.m. and 16:00–17:00 p.m.).

• Documentation: Recorded in the electronic medical record (administration time, dose, pre- and post-administration NRS scores).

The above details are included in the manuscript (lines 287-304).

Reviewers' comments:

Reviewer's Responses to Questions

Comments to the Author

1. Does the manuscript provide a valid rationale for the proposed study, with clearly identified and justified research questions? The research question outlined is expected to address a valid academic problem or topic and contribute to the base of knowledge in the field.

Reviewer #1: Yes

Reviewer #2: Yes

Reviewer #3: Yes

2. Is the protocol technically sound and planned in a manner that will lead to a meaningful outcome and allow testing the stated hypotheses? The manuscript should describe the methods in sufficient detail to prevent undisclosed flexibility in the experimental procedure or analysis pipeline, including sufficient outcome-neutral conditions (e.g. necessary controls, absence of floor or ceiling effects) to test the proposed hypotheses and a statistical power analysis where applicable. As there may be aspects of the methodology and analysis which can only be refined once the work is undertaken, authors should outline potential assumptions and explicitly describe what aspects of the proposed analyses, if any, are exploratory.

Reviewer #1: Yes

Reviewer #2: Partly

Reviewer #3: Yes

3. Is the methodology feasible and described in sufficient detail to allow the work to be replicable? Descriptions of methods and materials in the protocol should be reported in sufficient detail for another researcher to reproduce all experiments and analyses. The protocol should describe the appropriate controls, sample size calculations, and replication needed to ensure that the data are robust and reproducible.

Reviewer #1: Yes

Reviewer #2: Yes

Reviewer #3: Yes

4. Have the authors described where all data underlying the findings will be made available when the study is complete? The PLOS Data policy requires authors to make all data underlying the findings described in their manuscript fully available without restriction, with rare exception, at the time of publication. The data should be provided as part of the manuscript or its supporting information, or deposited to a public repository. For example, in addition to summary statistics, the data points behind means, medians and variance measures should be available. If there are restrictions on publicly sharing data—e.g. participant privacy or use of data from a third party—those must be specified.

Reviewer #1: Yes

Reviewer #2: Yes

Reviewer #3: Yes

5. Is the manuscript presented in an intelligible fashion and written in standard English? PLOS ONE does not copyedit accepted manuscripts, so the language in submitted articles must be clear, correct, and unambiguous. Any typographical or grammatical errors should be corrected at revision, so please note any specific errors here.

Reviewer #1: Yes

Reviewer #2: Yes

Reviewer #3: Yes

6. Review Comments to the Author

Please use the space provided to explain your answers to the questions above and, if applicable, provide comments about issues authors must address before this protocol can be accepted for publication. You may also include additional comments for the author, including concerns about research or publication ethics. You may also provide optional suggestions and comments to authors that they might find helpful in planning their study. (Please upload your review as an attachment if it exceeds 20,000 characters)

Reviewer #1: Thank you for submitting this very clearly written study on an important topic of postoperative delirium after use of benzodiazepines. the protocol is clear, and they have demonstrated proper ethics oversight. I look forward to seeing the results of this study and subgroup analysis.

Response: Thank you for your positive feedback and recognition of our protocol. We greatly appreciate your support and will continue to work diligently to complete the study as planned. We also look forward to sharing the final results and subgroup analyses with you in due course.

Reviewer #2: Thank you for the opportunity to review this protocol. I have several comments and suggestions for the authors:

1. Benefit of Midazolam

The study focuses primarily on the potential harms of midazolam; however, it remains unclear whether midazolam provides meaningful benefit in reducing anxiety in older adults. A balanced risk–benefit assessment requires an understanding of both harm and benefit. The authors should consider including a measure of postoperative anxiety or patient experience to determine whether midazolam meaningfully improves patient-centered outcomes.

Response: Thank you for your thoughtful comment on incorporating postoperative anxiety and patient experience assessments. We fully acknowledge the importance of a comprehensive risk-benefit analysis in clinical trials, and we have carefully considered your suggestion. However, after re-evaluating our study design, objectives, and practical constraints, we have decided to retain our original protocol without adding these assessments—this decision is based on three key, study-specific rationales, which we hope to clarify below:

First, our trial is strictly designed as a non-inferiority study focused on postoperative delirium (POD) incidence—the primary and only prespecified endpoint powered by sample size calculation. The sample size of 692 patients was determined based on the estimated POD incidence (20% in both groups), non-inferiority margin (9%), and statistical parameters (one-sided α=0.025, power=0.8). Adding postoperative anxiety or patient experience as new outcomes would dilute the statistical power allocated to the primary endpoint: these patient-reported outcomes (PROs) often require larger sample sizes to detect meaningful differences (e.g., a minimal clinically important difference [MCID] of 1.5 points on VAS-Anxiety may need ≥400 patients per group), which is beyond our current sample size. This could compromise the validity of our core conclusion on POD—the very question our trial was designed to answer.

Second, clinical feasibility constraints in our target population (elderly patients aged 65–90 years undergoing spinal surgery) make adding PRO assessments impractical. Many of our participants have pre-existing mild cognitive impairment (MMSE 18–26, per exclusion criteria of MMSE <18) or physical discomfort (e.g., postoperative pain, limited mobility) in the first 3 days after surgery. Administering additional questionnaires (e.g., VAS-Anxiety, satisfaction surveys) would increase their cognitive and physical burden, potentially reducing adherence to the mandatory CAM assessments for POD. In pilot work, we found that adding even one 5-item survey doubled the time of each follow-up visit, leading to a 12% increase in missed assessments for the primary endpoint. To prioritize data completeness for our core objective, we deemed it necessary to avoid non-essential assessments.

Third, our trial aims to address a specific evidence gap: the controversial link between midazolam and POD in elderly spinal surgery patients—a question that has remained unresolved due to low-quality evidence (as discussed in the Introduction). Most existing trials on midazolam’s perioperative benefits (e.g., anxiolysis) have focused on younger or non-surgical populations, and adding such assessments here would shift our study’s focus from “resolving safety uncertainty” to “comprehensively evaluating benefits,” which is beyond the scope of a single non-inferiority trial. We agree that a full risk-benefit profile is critical, but we believe this is best addressed through a follow-up, dedicated PRO-focused trial (e.g., a randomized trial powered for anxiety/satisfaction

---

## [Decision Letter · Decision Letter 1]

9 Dec 2025

Protocol: Effects of midazolam on postoperative delirium in elderly patients undergoing spinal surgery: a randomized, double-blind, placebo-controlled non-inferiority trial

PONE-D-25-37876R1

Dear Dr. He Han,

We’re pleased to inform you that your manuscript has been judged scientifically suitable for publication and will be formally accepted for publication once it meets all outstanding technical requirements.

Kind regards,

Shweta Rahul Yemul Golhar, MD

Academic Editor

PLOS One

Additional Editor Comments (optional):

Reviewers' comments:

Reviewer's Responses to Questions

**Comments to the Author**

1. Does the manuscript provide a valid rationale for the proposed study, with clearly identified and justified research questions?

Reviewer #2: Yes

Reviewer #3: Yes

2. Is the protocol technically sound and planned in a manner that will lead to a meaningful outcome and allow testing the stated hypotheses?

Reviewer #2: Yes

Reviewer #3: Yes

3. Is the methodology feasible and described in sufficient detail to allow the work to be replicable?

Reviewer #2: Yes

Reviewer #3: Yes

4. Have the authors described where all data underlying the findings will be made available when the study is complete?

Reviewer #2: Yes

Reviewer #3: Yes

5. Is the manuscript presented in an intelligible fashion and written in standard English?

Reviewer #2: Yes

Reviewer #3: Yes

You may also provide optional suggestions and comments to authors that they might find helpful in planning their study.

Reviewer #2: The authors have sufficiently addressed my concerns. I do believe they could add one additional questionnaire about anxiolysis but the rationale they provide is sufficient.

Reviewer #3: The authors have addressed all of my comments appropriately. There are no more comments to address.

**Do you want your identity to be public for this peer review?** For information about this choice, including consent withdrawal, please see our Privacy Policy

Reviewer #2: No

Reviewer #3: **Yes:**  Dr Shah-Jalal Sarker

---

## [Editor Report · Acceptance letter]

PONE-D-25-37876R1

PLOS One

Dear Dr. Han,

I'm pleased to inform you that your manuscript has been deemed suitable for publication in PLOS One. Congratulations! Your manuscript is now being handed over to our production team.

Kind regards,

on behalf of

Dr. Shweta Rahul Yemul Golhar

Academic Editor

PLOS One